# Overruling of Procalcitonin-Guided Antibiotics for Lower Respiratory Tract Infections in Primary Care: Ancillary Study of a Randomized Controlled Trial

**DOI:** 10.3390/antibiotics12020377

**Published:** 2023-02-12

**Authors:** José Knüsli, Loïc Lhopitallier, Andreas Kronenberg, Jean-Yves Meuwly, Onya Opota, Marc-Antoine Perrenoud, Marie-Anne Page, Kevin C. Kain, Aline Mamin, Valérie D’Acremont, Nicolas Senn, Yolanda Mueller, Isabella Locatelli, Noémie Boillat-Blanco

**Affiliations:** 1Infectious Diseases Service, Lausanne University Hospital, University of Lausanne, Rue du Bugnon 46, 1011 Lausanne, Switzerland; 2Gare10 Lausanne General Practice, Av. de la gare 10, 1003 Lausanne, Switzerland; 3Institute for Infectious Diseases, University of Bern, Friedbühlstrasse 51, 3001 Bern, Switzerland; 4Medix General Practice, Bubenbergplatz 8, 3011 Bern, Switzerland; 5Department of Radiology, Lausanne University Hospital, University of Lausanne, Rue du Bugnon 46, 1011 Lausanne, Switzerland; 6Microbiology Institute, Lausanne University Hospital, University of Lausanne, Rue du Bugnon 48, 1011 Lausanne, Switzerland; 7Tropical Disease Unit, Department of Medicine, University of Toronto, 200 Elizabeth Street, Toronto, ON M5G 2C4, Canada; 8Sandra Rotman Centre for Global Health, Toronto General Hospital, University Health Network, 105 St. George Street, Toronto, ON M5S 3E6, Canada; 9Division of Infectious Diseases and Centre for Emerging Viral Diseases, Faculty of Medicine, University of Geneva Hospitals, Rue Gabrielle-Perret-Gentil 4, 1211 Geneva, Switzerland; 10Digital Global Health Department, Centre for Primary Care and Public Health (Unisanté), University of Lausanne, Rue du Bugnon 44, 1011 Lausanne, Switzerland; 11Department of Family Medicine, Centre for Primary Care and Public Health (Unisanté), University of Lausanne, Rue du Bugnon 44, 1011 Lausanne, Switzerland; 12Department of Education, Research, and Innovation, Centre for Primary Care and Public Health (Unisanté), University of Lausanne, Rue du Bugnon 44, 1011 Lausanne, Switzerland

**Keywords:** antibiotic stewardship, procalcitonin, overruling, respiratory infection, primary health care

## Abstract

Background: Lower respiratory tract infections (LRTIs) in primary care are a promising target for antibiotic stewardship. A clinical trial in Switzerland showed a large decrease in antibiotic prescriptions with procalcitonin guidance (cut-off < 0.25 µg/L) compared with usual care. However, one-third of patients with low procalcitonin at baseline received antibiotics by day 28. Aim: To explore the factors associated with the overruling of initial procalcitonin guidance. Design and Setting: Secondary analysis of a cluster randomized trial in which patients with an LRTI were included. Method: Using the characteristics of patients, their disease, and general practitioners (GPs), we conducted a multivariate logistic regression, adjusted for clustering. Results: Ninety-five out of 301 (32%) patients with low procalcitonin received antibiotics by day 28. Factors associated with an overruling of procalcitonin guidance were: a history of chest pain (adjusted OR [aOR] 1.81, 95% confidence interval 1.03–3.17); a prescription of chest X-ray by the GP (aOR 4.65, 2.32–9.34); a C-reactive protein measured retrospectively above 100 mg/L (aOR 7.48, 2.34–23.93, reference ≤ 20 mg/L); the location of the GP practice in an urban setting (aOR 2.27, 1.18–4.37); and the GP’s number of years of experience (aOR per year 1.05, 1.01–1.09). Conclusions: Overruling of procalcitonin guidance was associated with GPs’ socio-demographic characteristics, pointing to the general behavioral problem of overprescription by physicians. Continuous medical education and communication training might support the successful implementation of procalcitonin point-of-care tests aimed at antibiotic stewardship.

## 1. Introduction

Estimates put the global toll of antimicrobial resistance at 1.27 million deaths in 2019, which underlines the need for urgent real-life implementation of antibiotic stewardship [1]. More than 80% of human antimicrobial prescriptions take place in primary care, and acute respiratory infections are the most common reason for unnecessary prescriptions, making them promising targets for stewardship [2,3]. Indeed, the majority of patients with lower respiratory tract infections (LRTIs) receive antibiotics, although only 5 to 10% of them have pneumonia in primary care and will benefit from this treatment [4,5].

Procalcitonin is a host biomarker that can differentiate bacterial from viral infections [6]. The recent availability of point-of-care tests enables general practitioners (GPs) to rapidly measure procalcitonin from capillary blood [7]. Procalcitonin use in primary care is not yet common in Switzerland, although its reimbursement by the Swiss health insurance system is currently under consideration. This might help it become a standard tool for the management of LRTIs. Indeed, a randomized clinical trial conducted in Swiss primary care showed that procalcitonin at the point of care to guide initiation of antibiotics (using a cut-off of <0.25 µg/L) for patients with LRTIs led to a large decrease in antibiotic prescription compared with usual care (−26%, 95% confidence interval [95% CI] −41% to −10%) [8]. However, although only 8% of patients had elevated procalcitonin at baseline and a recommendation for antibiotic prescription, 40% received antibiotics within 28 days of the initial consultation, suggesting room for improvement of the guidance. We aimed to identify factors associated with an overruling of the initial procalcitonin guidance that could support the real-life implementation of the procalcitonin point-of-care test.

## 2. Results

### 2.1. Participants

Figure 1 presents the flow of the participants. Of the 469 patients originally included in the randomized controlled trial, 329/469 (70%) were in the UltraPro or procalcitonin groups, had their procalcitonin measured at the point of care during the initial consultation, and their antibiotic prescription status assessed at day 28. Of those, the majority (301/329, 91%) had a procalcitonin value < 0.25 µg/L and were included in this ancillary study. Patients were mostly female (61%), 29% were 65 or older, 28% had comorbidities, 53% abnormal vital signs, 39% a bacterial or mixed infection and 7% a retrospectively measured CRP above 100 mg/l (see Table 1). Among these 301 patients with a procalcitonin value < 0.25 µg/L, 95 (32%) were prescribed antibiotics by day 28 (overruling of initial procalcitonin guidance), 26/95 (27%) during the initial consultation, 34/95 (36%) between 1 and 7 days after the initial consultation and 35/95 (37%) between 8 and 28 days after the initial consultation. Procalcitonin measurement was repeated during follow-up visits for 9/301 (3%) patients; the newly measured values were still < 0.25 µg/L.

### 2.2. Univariate Analysis

Patients in whom procalcitonin guidance was overruled had more often a history of chest pain (odds ratio [OR] 1.74, 95% CI 1.03–2.93), a temperature ≥ 37.8 °C (OR 2.16, 95% CI 1.04–4.48), a focal abnormal finding upon lung auscultation (OR 1.83, 95% CI 1.06–3.16), a prescription of chest X-ray from the GP (OR 3.79, 95% CI 1.85–7.76), a measurement of CRP by the GP during the consultation (OR 2.77, 95% CI 1.58–4.86), and a CRP value measured retrospectively (result not communicated to the GPs) above 100 mg/l (OR 4.59, 95% CI 1.52–13.90, reference CRP ≤ 20 mg/L) (Table 1). Among patients with CRP measured by the GP during the initial consultation (97/301, 33%), the CRP value was not associated with an overruling of procalcitonin guidance. The detection of bacteria was also not associated with an overruling.

### 2.3. Multivariate Analysis

The backward selection determined a model with 5 independent variables (Figure 2), some factors being characteristics of the patients and others of the GPs: a history of chest pain (adjusted OR [aOR] 1.81, 95% CI 1.03–3.17); the prescription of a chest X-ray by the GP (aOR 4.65, 95% CI 2.32–9.34); a CRP value measured retrospectively above 100 mg/L (aOR 7.48, 95% CI 2.34–23.93, reference CRP ≤ 20 mg/L); the location of the GP practice in an urban setting (aOR 2.27, 95% CI 1.18–4.37); and the GP’s number of years of experience (aOR per year of experience 1.05, 95% CI 1.01–1.09). The presence of comorbidities, abnormal vital signs or a bacterial etiology of the LRTI were not associated with an overruling of the procalcitonin guidance.

The intraclass correlation coefficient was 0. The Hosmer–Lemeshow test had a *p*-value of 0.44, meaning that good calibration of the model should not be rejected. The C statistic was 0.71, showing a reasonably good predictive value for the model.

## 3. Discussion

### 3.1. Summary

Our study, conducted on a patient population with an overall low severity of disease at presentation, shows that overruling the initial procalcitonin-based guidance in patients with LRTIs in primary care was associated with the characteristics of the patient (retrospectively measured CRP > 100 mg/L, history of chest pain) as well as of the GP (location of the practice in an urban setting, number of years of experience). The prescription of a chest radiograph by the GP was also strongly associated with an overruling of procalcitonin guidance.

Regarding the characteristics of the patients, a high retrospectively measured CRP result, which was not communicated to the GP, was associated with an overruling of procalcitonin guidance. However, the GP’s willingness to measure CRP and its results (available to the GP) did not influence the decision to prescribe antibiotics. Indeed, GPs measured the CRP in only 2 of the 18 patients (11%) with a retrospectively measured CRP > 100 mg/L. It has been previously shown that a CRP > 100 mg/L is associated with pneumonia, or even bacterial pneumonia [9,10]. However, in our study, the clinical outcome of patients was not related to CRP value; we found the same proportion of patients with a persistence of dyspnea at day 7 (98/258 (38%) versus 7/18 (41%) patients with CRP ≤ 100 mg/L and CRP > 100 mg/L, respectively, *p* = 1.0) and with an admission to the hospital (3/258 (1%) versus 1/18 (6%), respectively, *p* = 0.6), and this irrespective of antibiotic prescription. This discrepancy between procalcitonin and CRP values was rare and occurred in only 18/301 (7%) patients. Another patients’ characteristic associated with an overruling of procalcitonin guidance was the presence of chest pain. As chest pain has not been proven to be a predictor of pneumonia in several studies [11,12,13], we suggest that this argument alone should not justify a prescription of antibiotics. Our finding is not in line with another study in which chest pain was a protective factor against prescription [14].

The characteristics of the GPs associated with an overruling of procalcitonin guidance (urban setting and length of experience) point to other psychosocial determinants of antibiotic prescriptions. The multivariate analysis included the demographics, comorbidities and disease severity of the patients, suggesting that practicing in an urban area is associated with an overruling of procalcitonin guidance beyond differences in patient characteristics between settings, possibly due to differences in clinical practices. A study conducted in a neighboring country also found higher rates of antibiotic prescription in urban areas [15], while another one did not find any association [16]. Regarding the number of years of experience, the more experienced GPs might distrust the procalcitonin level in favor of their own clinical impression or might have retained more liberal prescribing patterns, predating our knowledge of antibiotic resistance. This finding is in line with previous studies showing that GPs who have practiced longer are more likely to prescribe antibiotics inappropriately [17,18]. Interestingly, in our dataset, GPs practicing in urban areas tended to be less experienced than their counterparts in rural regions (median number of years of experience: 8 versus 13 years in the urban versus rural group, respectively, *p* = 0.04). This might explain why the location of practice and the number of years of experience were not associated with the overruling of procalcitonin guidance in the univariate regression (two opposite factors canceling out), while the association became apparent in the multivariate model.

The last factor associated with an overruling of procalcitonin guidance is the prescription of a chest X-ray by the GP. This result was shared by an earlier study that found an increase in antibiotic prescriptions when physicians requested X-rays [19]. Of note, 60% (18/30) of X-rays were negative (as read by the GPs) among patients who received an antibiotic prescription. Presumably, GPs could reserve X-rays for patients with the perceived highest pre-test probability of pneumonia. Nevertheless, X-rays are useful only when their results are expected to change the management of patients.

Microbiology results obtained retrospectively were not associated with antibiotic prescriptions. This is in line with previous studies showing an absence of a correlation between the presence of bacteria and disease severity and course in primary care [20,21].

The results of this study underline the idiosyncrasies of medical care: clinical and biological arguments blend with psychosocial and relational considerations in the management of patients [22,23].

### 3.2. Strengths and Limitations

This study benefits from the strengths of the randomized controlled trial with a large number of practices included, the good quality and high completion of collected data, as well as the low rate of loss to follow-up. The biobank established during the trial made it possible to retrospectively analyze CRP and microbiology. We included several characteristics of the GPs in our variables, which proved to be associated with antibiotic prescriptions. Moreover, the calculated intraclass correlation coefficient was 0, suggesting that, conditional on the GP-dependent variables included in the model, no additional variation between GPs is estimated in the propensity to overrule the initial procalcitonin guidance. This lends another argument to the generalizability of our findings.

The main limitation of the study was the inability to differentiate between adequate and inadequate prescriptions. Consequently, no evidence-based improvement to the procalcitonin algorithm can be derived from our findings. This also means that our results neither support nor discourage the use of procalcitonin guidance beyond what was already proven in the original trial.

### 3.3. Comparison with Previous Studies

To the best of our knowledge, this is the first study to examine the factors associated with the overruling of procalcitonin-guided initiation of antibiotics for LRTIs in primary care. However, physicians explained the reasons for the overruling of procalcitonin guidance in two trials on procalcitonin-based management: one conducted in hospitals and one in primary care. Physicians justified overruling due to high clinical severity, respiratory or hemodynamic instability, signs of infection or when the patients requested antibiotics themselves. Physicians working in centers experienced with the use of procalcitonin were also more compliant with the recommendation [24,25].

A study on the associations between clinical presentation and antibiotic prescribing for acute exacerbation of chronic obstructive pulmonary disease in primary care, based on CRP measurement, showed that anomalies on lung auscultation were associated with higher odds of antibiotic prescription, while increased age and the presence of heart failure were associated with lower odds [26]. The auscultation and increased age did not appear in our multivariate model, although abnormal auscultation was indeed associated with higher odds of prescription in our univariate analysis; we had only a few patients (n = 0 in one group) with heart failure in our study, preventing us from using this variable in the analysis.

### 3.4. Implications for Research and Practice

Our results suggest that any future usage of procalcitonin guidance would probably benefit from new guidelines taking into account biomarker values for the management of LRTIs. This would help prevent uncertainty in the physicians’ assessment of patients when the procalcitonin values are at odds with X-ray results or CRP levels.

This study also demonstrated that the overruling of initial procalcitonin guidance was associated with GPs’ socio-demographic characteristics, pointing to the general behavioral problem of overprescription by physicians. A qualitative assessment of the drivers of antibiotic prescription might help identify the most adequate strategies and tools to support behavior change among GPs. In the meantime, continuous medical education on the use of procalcitonin and on “Less is More” approaches to prescribing, as well as communication training, might support the successful implementation of procalcitonin point-of-care tests aimed at antibiotic stewardship. This could lead to a further, clinically significant reduction in antibiotic prescriptions.

## 4. Materials & Methods

### 4.1. Study Design and Setting

This ancillary study is a secondary analysis of a three-group, open-label, cluster randomized trial in which 469 patients with LRTIs were included in Swiss primary care practices from September 2018 to March 2020 [8]. The detailed trial protocol has been published [27]. A total of 60 GPs were recruited, and their characteristics recorded. They were randomized to one of three study groups: antibiotics guided by sequential point-of-care procalcitonin and, in patients with elevated procalcitonin (≥0.25 µg/L), point-of-care lung ultrasonography (antibiotics recommended in patients with elevated procalcitonin and the presence of an infiltrate on lung ultrasound: UltraPro group); antibiotics guided by procalcitonin alone (antibiotics recommended in patients with elevated procalcitonin: procalcitonin group); or usual care. GPs could freely order additional diagnostic tests in all groups, as well as manage follow-up visits.

GPs included adult patients (aged 18 or older) presenting to their practice with an LRTI, defined as an acute cough and at least one of the following signs or symptoms: history of fever of more than 4 days, dyspnea, tachypnea (>22 cycles/min), or abnormal focal lung auscultation (i.e., definition of clinical pneumonia) [28]. Exclusion criteria were previously published [27]. Characteristics of the patients (demographics and comorbidities) and of their clinical presentation (symptoms and signs) as well as results of additional diagnostic tests ordered by the GPs outside the scope of the study protocol (chest X-ray, C-reactive protein [CRP]) were recorded by GPs in digitalized case-report forms using REDCap^®^ (Research Electronic Data Capture, Vanderbilt University, Tennessee). Patients had standardized telephone interviews on day 7 and day 28 after the initial consultation to assess if antibiotics had been prescribed. All participants gave their written informed consent. The trial was approved by the Swiss Ethics Committees of the cantons of Vaud and Bern (2017-01246).

### 4.2. Biological Samples and Retrospective Laboratory Analyses

During the initial consultation of patients from the intervention groups, plasma and nasopharyngeal samples were taken, stored at −20 °C within 6 h, and then moved to −80 °C within 72 h. Plasma was retrospectively analyzed for CRP using enzyme-linked immunosorbent assays (R & D DuoSet^®^, R & D Systems, Minnesota). Nasopharyngeal swabs were retrospectively analyzed for microbiological pathogens using two multiplex polymerase chain reaction assays: BioFire^®^ FilmArray^®^ Torch, Respiratory Panel 2.1 *plus* (detection of adenovirus 2/3/6/7/8, coronavirus 229E/HKU1/NL63/OC43/MERS-CoV/SARS-CoV-2, human metapneumovirus, human rhinovirus/enterovirus, influenza A including H1/H3/H1-2009 and B, parainfluenza virus 1/2/3/4, respiratory syncytial virus, *Bordetella parapertussis*, *Bordetella pertussis*, *Chlamydia pneumoniae*, *Mycoplasma pneumoniae*; BioFire Diagnostics, Salt Lake City, UT, USA) and Bacterial pneumonia CAP FTD-29 (detection of *Chlamydia pneumoniae*, *Haemophilus influenzae*, *Legionella pneumophila/longbeachae*, *Moraxella catarrhalis*, *Mycoplasma pneumoniae*, *Staphylococcus aureus*, *Streptococcus pneumoniae*; Fast Track Diagnostics, Esch-sur-Alzette, Luxembourg).

We defined a “bacterial or mixed infection” as the identification of at least one respiratory bacterium (cycle threshold < 35 for *Haemophilus influenzae*, *Moraxella catarrhalis*, *Streptococcus pneumoniae* and/or any positive result for *Chlamydia pneumoniae*, *Mycoplasma pneumoniae*, *Legionella pneumophila/longbeachae*), regardless of the identification of viruses, which made the most clinical sense for antibiotic stewardship. We defined a “viral infection” as the identification of at least one respiratory virus but no identification of bacteria.

The GPs were not aware of the CRP value or of the microbiological results.

### 4.3. Study Population and Outcome

In this secondary analysis, we included all patients of the intervention groups with a low procalcitonin value (<0.25 µg/L) as measured by a point-of-care test during the initial consultation. The outcome was the overruling of the initial procalcitonin-based guidance, defined as an antibiotic prescription by day 28, despite a low procalcitonin value at baseline.

### 4.4. Statistical Analyses

Descriptive statistics were reported as numbers and percentages for categorical variables or median and interquartile range for continuous variables.

We investigated factors associated with an overruling of procalcitonin guidance by GPs with univariate and multivariate mixed-effects logistic regression models, corrected for clustering within practices with a random effect, using backward selection with the likelihood ratio criterion. Variables included the characteristics of the patients and their clinical presentation (including vital signs with previously defined cut-offs: heart rate ≥ 100/min, respiratory rate > 22/min, systolic blood pressure ≤ 100 mmHg, temperature ≥ 37.8 °C) [27,29], their retrospectively measured CRP and microbiological results, as well as the characteristics of the GPs. The intraclass correlation of the multivariate mixed-effects logistic model was calculated using the π^2^/3 estimator of the within-cluster variation, i.e., the level-1 variance component of a logistic regression model [30]. Goodness of fit was assessed using the Hosmer–Lemeshow test and the C statistic (area under the receiver operating characteristic curve).

All statistical analyses were performed using R Statistical Software version 4.1.1. *p*-values < 0.05 were considered statistically significant.

## Figures and Tables

**Figure 1 antibiotics-12-00377-f001:**
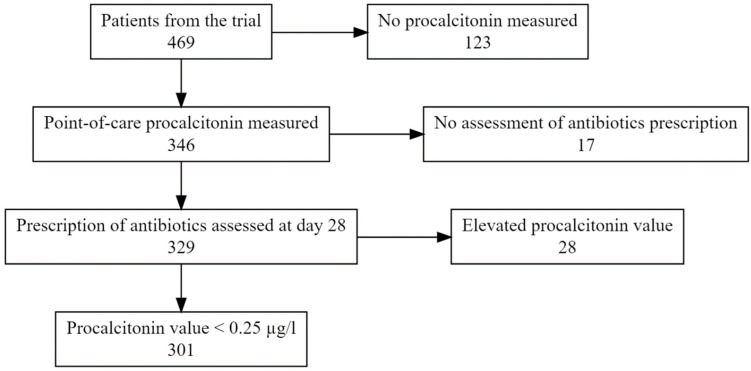
Participants’ flowchart.

**Figure 2 antibiotics-12-00377-f002:**
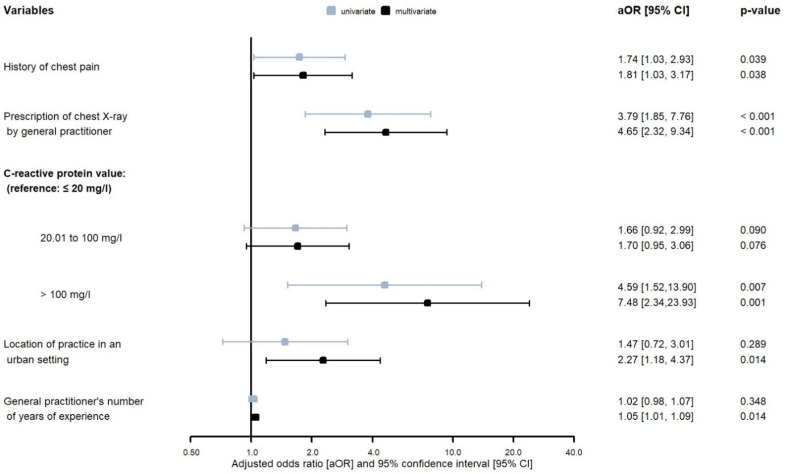
Forest plot showing factors associated with an overruling of initial procalcitonin guidance: univariate and multivariate logistic regression models corrected for clustering within practices.

**Table 1 antibiotics-12-00377-t001:** Patients’ characteristics, clinical presentation and general practitioners’ characteristics, according to the overruling of initial procalcitonin guidance: descriptive statistics and univariate logistic regression models. Values are n (% of the corresponding total number), unless stated otherwise. Odds ratios and *p*-values were calculated from univariate mixed-effects logistic regression and corrected for clustering.

Total Number	All Patients	No Overruling of Guidance	Overruling of Guidance	OR [95% CI]	*p*-Value
	N = 301	206 (68%)	95 (32%)		
**Demographics and comorbidities**					
Female	184 (61)	129 (63)	55 (58)	0.84 [0.50, 1.42]	0.52
Age ≥ 65 years	88 (29)	58 (28)	30 (32)	1.31 [0.74, 2.32]	0.35
Active smoker	62 (21)	38 (18)	24 (26)	1.47 [0.79, 2.74]	0.23
Any comorbidity among the following	84 (28)	63 (32)	21 (23)	0.65 [0.35, 1.19]	0.16
Diabetes	16 (5)	11 (5)	5 (5)	0.99 [0.31, 3.14]	0.99
Chronic obstructive pulmonary disease	17 (6)	12 (6)	5 (5)	0.96 [0.30, 3.00]	0.94
Asthma	52 (17)	41 (20)	11 (12)	0.52 [0.25, 1.12]	0.09
Heart failure	4 (1)	4 (2)	0 (0)	-	-
Other *	4 (1)	4 (2)	0 (0)	-	-
**Clinical presentation**					
History of fever	194 (65)	128 (62)	66 (70)	1.37 [0.78, 2.40]	0.27
History of dyspnea	205 (69)	144 (70)	61 (66)	0.84 [0.47, 1.48]	0.54
History of chest pain	140 (47)	87 (43)	53 (56)	1.74 [1.03, 2.93]	0.04
History of sputum production	209 (70)	143 (70)	66 (70)	1.06 [0.60, 1.87]	0.85
Heart rate ≥ 100/min	47 (16)	30 (15)	17 (18)	1.29 [0.64, 2.58]	0.48
Respiratory rate > 22/min	49 (16)	36 (18)	13 (14)	0.87 [0.40, 1.87]	0.72
Systolic blood pressure ≤ 100 mmHg	15 (5)	7 (3)	8 (9)	2.21 [0.70, 6.99]	0.18
Temperature ≥ 37.8 °C	44 (15)	25 (12)	19 (20)	2.16 [1.04, 4.48]	0.04
Focal abnormal finding upon lung auscultation	137 (46)	86 (42)	51 (54)	1.83 [1.06, 3.16]	0.03
**Management**					
Prescription of chest X-ray by GP	56 (19)	26 (13)	30 (32)	3.79 [1.85, 7.76]	<0.001
**Biomarkers**					
CRP measured by GP	97 (33)	51 (25)	46 (49)	2.77 [1.58, 4.86]	<0.001
CRP value (measured retrospectively)					
≤20 mg/L	159 (58)	116 (62)	43 (48)	ref	ref
20.01–100 mg/L	99 (36)	64 (34)	35 (39)	1.66 [0.92, 2.99]	0.09
>100 mg/L	18 (7)	7 (4)	11 (12)	4.59 [1.52, 13.90]	0.007
**Microbiology**					
Viral infection	117 (41)	79 (40)	38 (43)	ref	ref
Bacterial or mixed infection	112 (39)	75 (38)	37 (42)	1.19 [0.65, 2.19]	0.58
No pathogen identified	56 (20)	42 (21)	14 (16)	0.75 [0.35, 1.60]	0.45
**General practitioner’s characteristics**					
Francophone	274 (91)	190 (92)	84 (88)	0.66 [0.22, 1.96]	0.45
>5 general practitioners in practice	89 (30)	57 (28)	32 (34)	1.37 [0.64, 2.94]	0.42
Location of practice in an urban setting	187 (62)	121 (59)	66 (70)	1.47 [0.72, 3.01]	0.29
Years of experience in practice (median [IQR])	9.0 [5.0, 19]	8.0 [5.0, 19]	10 [5.0, 19]	1.02 [0.98, 1.07]	0.35

OR: Odds ratio. 95% CI: 95% confidence interval. GP: General practitioner. CRP: C-reactive protein. IQR: Interquartile range. ref: Reference value for logistic regression. *: Active cancer, chronic kidney disease, or human immunodeficiency virus infection.

## Data Availability

The original UltraPro trial data are publicly available in the Zenodo repository at https://doi.org/10.5281/zenodo.4032527 (accessed on 20 January 2023). The microbiological data used in this study are available on request from the corresponding author; the data are not publicly available due to work still in progress on the microbiological samples database.

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
