# Peer review of "Overruling of Procalcitonin-Guided Antibiotics for Lower Respiratory Tract Infections in Primary Care: Ancillary Study of a Randomized Controlled Trial"

_antibiotics, 2023, doi:10.3390/antibiotics12020377_

Round 1

Reviewer 1 Report

This study investigates the parameters associated with the overriding of procalcitonin-guided antibiotic initiation for LRTIs. Regarding routine clinical practice in some places, this is a really intriguing topic.

comment

1. The introduction should provide additional information about the routine use of procalcitonin in standard care to emphasize the significance of this study.

2. Should add additional discussion about the future usage of procalcitonin guidance when determining factors that override procalcitonin guidance and whether or not these data support the continued use of procalcitonin guidance in clinical practice.

Author Response

Dear reviewer,

Thank you for your comments.

  • I have added a sentence about the use of procalcitonin in primary care in the Introduction, which indeed gives the study more context.
  • Unfortunately, as the study was not designed to evaluate if the prescriptions were adequate or not, we are unable to support or discourage the use of procalcitonin from its results, which I detailed in the "Limitations" section of the Discussion.
  • Finally, I have added a paragraph on the future use of procalcitonin in the last section, which I hope will help clarify what we think is necessary next to make procalcitonin relevant in the future management of LRTIs, considering our results.

I hope I was able to adequately answer your points with those revisions.

Best regards,

José Knüsli

Reviewer 2 Report

A manuscript that identifies the determinants of the prescription of antibiotics in peripheral airway infections beyond what was foreseen by a protocol based on a fixed value of procalcitonin. Five determinants of this overprescribing have been identified and the authors underline that some of these are characteristic of the doctor: years of activity and location of the practice in an urban environment.Those results, although not entirely surprising, are interesting as a further demonstration of the interaction between a decision-making algorithm and what we may call the human factor. Whether this interaction was positive or negative is outside the design of the study, but we know that, in many situations governed by decision-making algorithms, the operator is left with the option of superimposing his decision on that of the algorithm, and in this way increasing the probability of a positive result. 

I have no major objections to this manuscript but I believe that a less negative notation of the human factor would be more consistent with the results of the study. The deletion of the term "mostly" in line 288 may put the conclusions in a much better  agreement with the odds ratio values ​ reported in Figure 2.

Finally, the odds ratio value associated with the urban location of the practice (probably the only one not plausibly linked to an improvement in the results of the therapy) shows a significant increase of value between univariate and multivariate analyses. I think a comment on this point, with respect to the method used to define the multivariate model, would be appropriate.

Author Response

Dear reviewer,

Thank you for your comments.

  • Indeed, I had used the term "mostly" in terms of raw numbers of associated factors, but if we look at the odds ratios, the CRP value has the highest association. My wording was indeed ambiguous and I have removed it in this revised version.
  • Your comment about the urban location of the practice becoming significantly associated with the overruling of procalcitonin guidance in the multivariate analysis was very interesting. I realized that the same was true for the number of years of experience, and wondered if those two factors had to be corrected by each other in the logistic regression for their effect to become apparent, my unproven hypothesis being that maybe rural regions of Switzerland tend to have "older" or more experienced practitioners, whereas urban areas have "younger" or less experienced GPs (which is actually a known problem of Swiss rural regions, since retiring GPs there have trouble finding younger GPs to take their place). So I checked in our dataset and this was indeed the case for our GPs, which could explain why the effect of those 2 factors could have canceled each other in the univariate analysis, which I detailed in the revised Discussion. However, I was unable to link this to the backward selection we used to determine the multivariate model. The backward selection algorithm eliminated factors one by one by comparing the multivariate models' goodness of fit (likelihood ratio criterion) and this algorithm selected (among others) the location of the practice and the number of years of experience of the GPs as giving the best (in the goodness of fit sense) multivariate model. The fact that they are, in this multivariate model, indeed associated with the overruling of procalcitonin guidance indeed gives them a lot more probability to be selected by the algorithm, but I do not think this is dependent on their association in the univariate models.

I hope I was able to adequately answer your points with those revisions.

Best regards,

José Knüsli